# Short-Term Exposure to High Atmospheric Vapor Pressure Deficit (VPD) Severely Impacts Durum Wheat Carbon and Nitrogen Metabolism in the Absence of Edaphic Water Stress

**DOI:** 10.3390/plants10010120

**Published:** 2021-01-08

**Authors:** Dorra Fakhet, Fermín Morales, Iván Jauregui, Gorka Erice, Pedro M. Aparicio-Tejo, Carmen González-Murua, Ricardo Aroca, Juan J. Irigoyen, Iker Aranjuelo

**Affiliations:** 1Instituto de Agrobiotecnología (IdAB), Consejo Superior de Investigaciones Científicas (CSIC)-Gobierno de Navarra, Avenida Pamplona 123, 31192 Mutilva, Spain; dorra.fakhet@csic.es (D.F.); fmorales@eead.csic.es (F.M.); 2Departamento Ciencias del Medio Natural, Campus de Arrosadía, Universidad Pública de Navarra, 31192 Pamplona, Spain; ivan.jauregui@uliege.be (I.J.); pmapariciotejo@unavarra.es (P.M.A.-T.); 3Atens, Agrotecnologías Naturales S.L., La Riera de Gaia, 43762 Tarragona, Spain; gorka.erice@atens.es; 4Departamento de Biología Vegetal y Ecología, Facultad de Ciencia y Tecnología, Universidad del País Vasco/Euskal Herriko Unibertsitatea (UPV/EHU), Barrio Sarriena s/n, 48940 Leioa-Bizkaia, Spain; carmen.gmurua@ehu.eus; 5Departamento de Microbiología del Suelo y Sistemas Simbióticos, Estación Experimental del Zaidín (CSIC), Profesor Albareda 1, 18008 Granada, Spain; Ricardo.aroca@eez.csic.es; 6Plant Stress Physiology Group, Associated Unit to CSIC (EEAD, Zaragoza and ICVV, Logroño), Faculty of Sciences, Universidad de Navarra, Irunlarrea, 1, 31008 Pamplona, Spain; jirigo@unav.es

**Keywords:** C/N metabolism, drought, durum wheat, physiology, relative humidity, vapor pressure deficit

## Abstract

Low atmospheric relative humidity (RH) accompanied by elevated air temperature and decreased precipitation are environmental challenges that wheat production will face in future decades. These changes to the atmosphere are causing increases in air vapor pressure deficit (VPD) and low soil water availability during certain periods of the wheat-growing season. The main objective of this study was to analyze the physiological, metabolic, and transcriptional response of carbon (C) and nitrogen (N) metabolism of wheat (*Triticum durum* cv. Sula) to increases in VPD and soil water stress conditions, either alone or in combination. Plants were first grown in well-watered conditions and near-ambient temperature and RH in temperature-gradient greenhouses until anthesis, and they were then subjected to two different water regimes well-watered (WW) and water-stressed (WS), i.e., watered at 50% of the control for one week, followed by two VPD levels (low, 1.01/0.36 KPa and high, 2.27/0.62 KPa; day/night) for five additional days. Both VPD and soil water content had an important impact on water status and the plant physiological apparatus. While high VPD and water stress-induced stomatal closure affected photosynthetic rates, in the case of plants watered at 50%, high VPD also caused a direct impairment of the RuBisCO large subunit, RuBisCO activase and the electron transport rate. Regarding N metabolism, the gene expression, nitrite reductase (NIR) and transport levels detected in young leaves, as well as determinations of the δ^15^N and amino acid profiles (arginine, leucine, tryptophan, aspartic acid, and serine) indicated activation of N metabolism and final transport of nitrate to leaves and photosynthesizing cells. On the other hand, under low VPD conditions, a positive effect was only observed on gene expression related to the final step of nitrate supply to photosynthesizing cells, whereas the amount of ^15^N supplied to the roots that reached the leaves decreased. Such an effect would suggest an impaired N remobilization from other organs to young leaves under water stress conditions and low VPD.

## 1. Introduction

Increased temperature, elevated CO_2_ concentrations, and reduced rainfall are the main characteristics of future climate projections. Global warming has been increasing by 0.2 °C per decade, and predictions of the total increase in global mean temperatures indicate a mean of 1.5 °C being reached between 2030 and 2052 if the frequency of extreme weather events continues to increase at the current rate [1]. Increases in air temperature are accompanied by decreases in atmospheric relative humidity (RH), leading to changes in evaporative demand, also known as vapor pressure deficit (VPD). Small increases in temperature and/or decrements in RH cause drastic changes in VPD. Water movement in plants is tightly conditioned by the difference between the VPD in the stomatal cavities and the atmospheric VPD. A recent publication by Yuan et al. [2] described the global trends in the vegetative index (a proxy for plant photosynthesis) in response to increases in VPD since 1990, highlighting the relationship of forest mortality to higher VPD. Lobel and coworkers [3] also identified the hidden effect of VPD in decreasing corn yield in recent decades, and their model simulations suggest even more severe reductions in the future in the corn belt in the United States. These are just a few examples of how VPD drives global ecosystems and plant photosynthesis.

Durum wheat is one of the most widely cultivated crops in the Mediterranean basin, a region that is classified as a “climate risk hotspot” [4]. In the coming years, Mediterranean staple crops, such as wheat, will face new environmental challenges including extreme dry and warm seasons, exposure to high VPD (driven by elevated temperatures and low RH), and a pronounced decrease in precipitation [5] during the different plant phenological stages, among which grain filling has critical importance. Studies from all over the world dealing with the effect of climate change on staple crop growth [6,7] have revealed the susceptibility of modern seed varieties to these changes [8]. Handling the impact of these changes on crop production will need very careful management of water, nutrients, and crop genetic potential.

Contrasting VPD has a considerable impact on plant carbon (C) metabolism. In fact, both diffusional (stomatal and mesophyll conductance, g_s_ and g_m_) and biochemical limitations to photosynthesis have been reported in response to increases in VPD [9,10,11,12,13,14]. Moreover, elevated VPD can reduce photosynthesis in the whole cereal plant [14]. Lihavainen and coworkers [15] have reported a metabolic adjustment in response to low VPD that modified N allocation and reflected N deficiency. In fact, further information on the impact of contrasting VPD on N metabolism is scarce, however, in some studies, the role of NRT genes (Nitrogen Transporter Genes) was investigated in drought conditions in bread wheat [16]. Other NRT genes were characterized in *Arabidopsis thaliana* [17,18]. Dechorgnat et al. [19] reported a schematic representation of nitrate routes within the *Arabidopsis* plant.

Plant responses to drought vary from adaptive to deleterious, depending on the intensity and the duration of water scarcity [20,21]. The damage caused by soil water shortage during plant growth on plant metabolism has been the subject of many studies. A severe water deficit during the first plant growth stages has a significant, negative impact on biomass production and yield [22,23,24]. When soil water deficit occurs during wheat plant maturation, accelerated senescence of the whole plant is observed [25]. This is a particularly notable feature of the flag leaf, and it serves as a key influence on wheat yield [26]. During this process, chlorophyll and N decrease, and pre-anthesis photo-assimilates and nutrients (N) are remobilized from source tissues (leaves) to sink tissues (grain) and contribute to grain filling [27,28].

Drought tolerant genotypes tend to show resilience to high VPD. Within this context, the objective of our study was to investigate the performance of the drought-tolerant durum wheat cultivar Sula in response to different atmospheric VPD and substrate water availabilities during grain filling, through the analysis of different parameters dealing with carbon and nitrogen assimilation.

Thus, some C and N metabolism components were selected and studied in flag leaf, using (i) gas exchange measurements, (ii) RuBisCO large subunit and RuBisCO activase gene expression, (iii) ^13^C and ^18^O isotopic composition together with ^15^N labeling approaches, (iv) N uptake and long-distance transport (nitrate transporters gene expression, glutamine synthetase, nitrite reductase (NIR), and (v) amino acid composition.

## 2. Results

Three situations occurring frequently under field conditions were simulated with our greenhouse experiments: (i) well-watered plants that suffer a heatwave (with high VPDs for several days), (ii) plants under drought that suddenly have cooler days and higher RHs in which the VPD decreases markedly, and (iii) plants under drought with high VPDs (the typical situation in summer in the Mediterranean region) (Figure 1).

### 2.1. Gas Exchange Measurements

Exposure to low VPD produced remarkably higher net photosynthetic rate (A_N_) values in the flag leaf of durum wheat plants under both of the water availability regimes (Figure 2a). The lowest and highest A_N_ values were found in the water-stressed plants subjected to high VPD and under well-watered and low VPD conditions respectively, whereas the rest of the treatments showed intermediate values (Figure 2a). This impaired photosynthesis caused by VPD was accompanied by stomatal closure (decreased g_s_) in well-watered plants alone, whereas no significant change was observed under water stress conditions for both VPD levels (Figure 2b). Stomatal conductance likely did not decrease significantly with increased VPD in water-stressed plants because stomata were already more closed than in the well-watered ones (Figure 2b). However, VPD caused no differences in the intrinsic transpiration rate (E), whereas water availability did, and it largely reduced E in plants exposed to limited water availability to the same degree for both VPD levels (Figure 2c). As a consequence of CO_2_ exchanged with the atmosphere and fixed by the plants, the sub-stomatal CO_2_ concentration (Ci) remained fairly constant irrespective of the treatments (Figure 2d). Maximum carboxylation velocity of Rubisco (Vcmax) had a behavior similar to A_N_ with the lowest values in water-stressed plants at high VPD (Figure 2e). Interestingly, the exposure to low VPD tended to increase the electron transport rate (Jmax), while the water availability treatment did not in the water-deficient treatment plants exposed to low VPD, which mostly showed high Jmax values, while a similar but non-significant response was found in well-watered conditions (Figure 2f). No remarkable changes were observed in the Vcmax/Jmax ratio (Figure 2g).

### 2.2. Stable Isotope Signatures

#### 2.2.1. ^18^O Isotopic Composition, δ^18^O

Results of the δ^18^O measured in the total organic matter (TOM) of flag leaves indicated that under well-watered conditions and high VPD, leaf tissues were considerably enriched with the O heavy isotope ^18^O relative to leaves exposed to low VPD (Figure 3a). Under water stress, however, there were no statistical differences in response to different VPDs (Figure 3a). Additionally, leaves from plants grown in well-watered conditions under low VPD discriminated in a similar way to those grown in water-stressed substrates under low VPD (Figure 3a).

#### 2.2.2. ^15^N Isotopic Composition, δ^15^N

Under full irrigation, high VPD seemed to activate N transport when compared to plants exposed to low VPD. Nitrogen isotopic composition was measured in flag leaves of plants grown with both non-labeled substrates (Figure 3b) and those labeled with ^15^N (Figure 3c). Under well-watered conditions, the ^15^N that reached flag leaf tissues significantly increased at low VPD (Figure 3c). Under water stress conditions, the ^15^N in the flag leaf was low under both VPD levels (Figure 3c). Interestingly, the results of natural variation in ^15^N point was in the same trend as the ^15^N labeled, but the large variation prevented observation of significant differences (Figure 3b).

#### 2.2.3. ^13^C Isotopic Composition, δ^13^C

Carbon isotopic composition was measured in flag leaves in both the TOM (Figure 3d) and water-soluble compounds (WSCs) (Figure 3e). Under well-watered conditions, high VPD treatment enriched leaves in ^13^C with respect to those under low VPD, which was better reflected in the recently fixed C of the WSCs (with decreases in δ^13^C) than in the TOM where the decreases in δ^13^C were not statistically significant.

Under water stress conditions, different atmospheric VPDs did not result in changes in flag leaf ^13^C composition measured either in the TOM (Figure 3d) or WSCs (Figure 3e).

### 2.3. Gene Expression

#### 2.3.1. RuBisCO Synthesis and RuBisCO Activase Genes

The expression levels of the genes involved in the synthesis of the RuBisCO large subunit (*RBCL*) (Figure 4a) and the RuBisCO activase β subunit (*RCAB*) (Figure 4b) were investigated. Significant effects were observed in response to increases in VPD under well-watered conditions or water-stress conditions (Figure 4a,b). RuBisCO synthesis (Figure 4a) and activase (Figure 4b) gene expression was significantly reduced by water stress. In addition, results showed a tendency for *RBCL* gene expression to increase in plants exposed to water stress and low VPD and this trend was significant (Figure 4a). On the other hand, *RCAB* gene expression in plants exposed to water stress and low VPD increased significantly compared to those exposed to water stress and high VPD (Figure 4b).

#### 2.3.2. Nitrogen Assimilation and Transport Genes

##### Nitrate Transport Genes

Transcription levels of the different genes involved in nitrate transport within the different plant parts were analyzed. The transcription level of *NRT1.3*, which is the gene proposed to be involved in nitrate transport in photosynthesizing cells, was observed to be the lowest in water-stressed plants with high atmospheric VPD, followed by well-watered plants (both at low and high VPD levels), whereas water-stressed plants with low atmospheric VPD had the highest *NRT1.3* gene expression (Figure 4c). From these results, it can be suggested that nitrate transport in parenchymal cells in water-stressed roots improves when the atmospheric conditions change from high to low VPD. The *NRT1.5* gene codes for a nitrate transporter loading from root tissue into the xylem. With the exception of well-watered plants at low VPD, the expression of the *NRT1.5* gene was not affected by either the changes in soil water conditions or the atmospheric VPD levels (Figure 4d). *NRT1.8* gene encodes a nitrate transporter that facilitates the unloading of nitrate from the xylem, which is the first step in the delivery of nitrate to different plant organs. The expression of this gene was decreased by water stress, but there was no effect of VPD on gene expression in water-stressed plants (Figure 4e). Flag leaves are N sinks, where N is used in a large variety of anabolic reactions. The *NRT1.4* gene encodes a nitrate transporter that facilitates the distribution of nitrate from the xylem towards leaves. A high VPD under well-watered conditions resulted in high expression levels of this gene, whereas water stress (irrespective of the atmospheric VPD) led to low expression of the *NRT1.4* gene (Figure 4f). Furthermore, gene expression, encoding for nitrate transporters that facilitate remobilization from old, senescent leaves to growing shoots and seeds, and unloading from phloem to roots, was significantly affected by soil water conditions and the atmospheric VPD levels in most cases, showing very contrasting results. In fact, the expression levels of the *NRT1.7* (Figure 4g) and *NRT1.9* (Figure 4h) genes, which achieve these functions, respectively, decreased from high to low VPD in well-irrigated plants but increased from high to low VPD in water-stressed plants. Thus, among the above-mentioned nitrate assimilation steps, nitrate distribution towards flag leaves and nitrate transport to mesophyll cells seemed to be the phases that were most affected by water deficiency and atmospheric VPD levels (based on these nitrate transporters gene expression results).

#### 2.3.3. Nitrogen Assimilation 

Nitrogen assimilation was studied in this work through the analysis of the transcription level of the *GS2* gene, coding for glutamine synthetase, an enzyme that converts glutamate to glutamine, and the NIR gene, which codes for the synthesis of the NIR enzyme. Results show that *GS2* gene expression was negatively affected by soil water deficiency. In fact, the transcription levels of the *GS2* gene were lower in water-stressed plants than in well-watered plants (Figure 4i). Moreover, the gene encoding NIR was down-regulated by soil water deficiency under high VPD levels but up-regulated under low VPD levels (Figure 4j). For both genes, the highest transcript levels were observed in well-watered plants and under high VPD (Figure 4i,j).

### 2.4. Amino Acid Concentrations

To obtain a complete characterization of plant N metabolism in changing conditions of soil moisture and atmospheric VPD levels, amino acid concentrations were analyzed in flag leaves for the four treatments (well-watered/high VPD, low VPD, water-stressed/high VPD, low VPD) (Table 1). Amino acids such as GLY and THR were affected by water availability, whereas the concentration of GLU was not affected, irrespective of the VPD conditions (Table 1). In addition, the concentrations of amino acids including ARG, LYS, LEU, ILE, MET, PHE, TRP, HIS, TYR, ASP, VAL, GLN, PRO, SER, ALA, and ASN were affected by water stress and/or the VPD level (Table 1). In well-watered conditions, the low VPD caused a considerable decrease in the flag leaf concentrations of ARG, TRP, SER, LEU, VAL, PRO, and GLN. In contrast, water stress increased the concentrations of these amino acids. Amino acids such as LYS, ILE, MET, PHE, and TYR were more influenced by the water regime than the atmospheric VPD, with significant increases in concentrations under water-deficient conditions but smaller, although significant differences between the VPD levels. However, ALA, demonstrated different behavior, being less concentrated under water stress and low VPD than in the other three treatments. 

## 3. Discussion

Crops in the field face both biotic and abiotic stress factors. How abiotic stress manifests can be very different depending on environmental attributes such as soil water availability and atmospheric conditions: (i) well-watered plants that suffer a heatwave (with high VPDs for several days), (ii) plants under drought that suddenly have cooler days and higher RHs in which the VPD decreases markedly, and (iii) plants under drought with high VPDs (the typical situation in summer in the Mediterranean region). All three of these situations can be simulated/tested with greenhouse experiments, modifying the temperature and/or the RH in order to change the VPD. In the experiments reported here, we have modified both RH and temperature to increase or decrease the VPD, using well-irrigated plants with high RH and slightly lower temperature exposure as control plants. Thus, durum wheat plants were first grown in temperature-gradient greenhouses until anthesis under full irrigation and near-ambient temperature and RH, and treatments then started by subjecting plants to two different water regimes (full irrigation and irrigation at 50%) over one week, and finally exposure to two VPD levels (low and high) for five additional days. Specific carbon and N metabolism components were selected and investigated due to their target role on grain yield. Previous studies have pointed towards a tight correlation between stomatal aperture (and consequently transpiration rate) and plant water uptake, which as a side effect influences the amount of nutrients (including N) that are absorbed by the root [15]. The results indicated that a high VPD in well-irrigated plants impacted many of the C metabolism components and activated also some other N metabolism aspects in the durum wheat flag leaf.

When plants are well watered, high atmospheric VPD had significant effects on durum wheat photosynthesis and carbon assimilation and also on nitrogen transporter gene expression (Figure 5a). In fact, at elevated temperature and low RH (high VPD), a significant decrease in stomatal conductance was observed, resulting in lowered photosynthetic rates, which impacts carbon assimilation. It has been reported among different species that low atmospheric RH and good soil water conditions are associated with decreases in photosynthetic rates and stomatal conductance when leaf-air VPD exceeds a certain threshold [11,29,30,31,32]. Under elevated air temperatures, photosynthetic capacity has been observed to decrease [33] and is limited by a decline in RuBP regeneration, RuBisCO carboxylation activity, or triose phosphate utilization [12,30,34]. Regarding N metabolism, gene expression related to mesophyll N uptake (by the nitrate transporter gene *NRT1.3*; [17]), NIR and N transport to young leaves (the *NRT1.4* gene; [19]), the amount of ^15^N supplied to the roots reaching the leaves, and the concentration of amino acids in leaves (ARG, LEU, TRP, ASP, GLN, and SER) increased under high VPD in well-watered conditions, and this suggested activation of N metabolism and final transport to the leaves and photosynthesizing cells. Glutamine and ASP have been identified as regulators of nitrate reductase gene expression [35], whereas the *NIR* gene seems to be controlled photosynthetically [36]. The accumulation of ARG, LEU, TRP, ASP, GLN, and SER under high atmospheric VPD in well-irrigated plants implicates these amino acids in the plant response to dry air conditions. This situation simulates plants in the field suffering a heatwave; there is enough soil water but the high atmospheric demand (due to low RH and high temperature) affects leaf transpiration [3,10,37,38] and leaf metabolism as described here. These are novel results with respect to N metabolism.

In terms of carbon assimilation and photosynthesis, water stress resulted in decreases in stomatal conductance, photosynthetic electron transport, and RuBisCO large subunit and RuBisCO activase gene expression, all of which lowered photosynthetic rates (Figure 5b). Farquhar and coworkers [39] explained the relationship between stomatal conductance and CO_2_ assimilation, highlighting the role of stomata in providing the required amount of CO_2_ for photosynthesis in the chloroplast [20,40]. Nevertheless, C fixation is not only limited by diffusional processes; leaf biochemistry also plays an important role [41,42,43,44,45]. With regards to nitrogen uptake and assimilation, the occurrence of an almost generalized increase in the amino acids without an increase in gene expression related to root N uptake (Figure 5b) as well as impaired N remobilization from roots to young leaves (xylem unloading analyzed by gene expression of the nitrate transporter *NRT1.8*; [46]), both suggested protein degradation. The accumulation of PRO observed in water-stressed plants (Table 1) was previously reported as an indicator of soil water shortage conditions [47] and cellular osmoprotectant [48,49]. These experimental conditions resemble plants in the field in summer, with low water availability in the soil and high atmospheric VPD.

On the other hand, it should be noted that the response of plants to water shortage may or may not be reversible and depends not only on the intensity of the drought but also the duration of the applied stress. Carmer and coworkers [50] reported that high atmospheric RH (low VPD) promotes stomatal opening, and consequently increases stomatal conductance and CO_2_ diffusion into the mesophyll. In the present work, the results showed that under such dry substrate conditions, a decrease in VPD ameliorated the effect of drought on C metabolism, increasing photosynthesis (mostly related to increases in photosynthetic electron transport rates) (Figure 5c). Regarding nitrogen uptake and transport, McDonald et al. [51] observed that a high atmospheric RH (low leaf-air VPD) decreased both the transpiration rate and the N^15^ content, which the authors attributed to an impaired N uptake. However, our data indicated a large, positive effect on gene expression related to mesophyll cell N uptake, but with decreases in the amount of ^15^N supplied to the roots reaching the leaves, which points instead to an impaired, low VPD-mediated, N remobilization from roots to flag leaves in water-stressed plants.

## 4. Materials and Methods

### 4.1. Experimental Setup

The experiment was conducted with a modern (semidwarf) non-waxy Spanish variety of durum wheat (*Triticum durum* Desf. cv. Sula). The modern durum wheat cultivar Sula is one the most widely cultivated in the Mediterranean region, showing good yield and quality grain under elevated CO_2_ conditions [52].

Seeds were vernalized for 4 weeks at 4 °C. Seedlings were later transplanted to 3 L plastic pots (4 plants/pot) containing a mixture of vermiculite, perlite, and peat (2:2:1, *v*/*v*/*v*) in a greenhouse located at the University of Navarre [53] in Pamplona (Spain). A total of 32 plants was grown with an average temperature of 25/18 °C day/night, and relative humidity between 60% and 80%. Plants were watered with a complete Hoagland solution twice a week (using NO_3_ as the N source), and to avoid salt accumulation, water was added to the plants once a week. When the plants reached the anthesis stage (the Zadoks scale, Z_65_), they were transferred to controlled growth chambers. The plants were grown 4 days with a day/night temperature and RH of 25/18 °C and 60/80% (a VPD of 1.27/0.42 KPa, day/night), respectively, and a 14 h photoperiod. The illumination was based on halogen and fluorescent lamps, reaching a photosynthetic photon flux density (PPFD) of 400 µmol m^−2^ s^−1^ at the plant level. Once the plants were acclimated to conditions in the chambers, they were separated into two groups according to two water regimes: well-watered (WW) plants that were watered every day, and water-stressed (WS) plants. WS plants were irrigated with 50% of the water supplied to the control, WW plants. After one week, WW and WS plants were further divided into two additional groups according to the ambient vapor pressure deficit (VPD) treatments (Figure 1).

In each water treatment, plants were exposed to different RH and temperature levels creating a difference in the VPD. The high VPD level (2.27/0.62 KPa, day/night) was obtained when the growth chamber temperature was maintained at 28/18 °C (day/night) and at RH levels of 40/70% (day/night). On the other hand, the low VPD level conditions (1.01/0.36 KPa, day/night) were obtained when the growth temperature was at 26/16 °C (day/night) and the RH levels were at 70/80% (day/night). After 5 days of treatment, the gas exchange measurements and sampling were conducted on flag leaves. Determinations were carried out in flag leaf because of its significant contribution to the total final yield in wheat.

### 4.2. Gas Exchange

Healthy flag leaves were chosen to conduct gas exchange measurements 2 h after the start of illumination in the growth chambers, using a Li-Cor 6400XT portable photosynthesis system (Li-Cor, Lincoln, NE, USA). The gas exchange response to [CO_2_] was measured by changing the [CO_2_] entering the leaf chamber with the following steps: 400, 300, 250, 200, 150, 100, 50, 400, 500, 600, 700, 800, 1000, 1200, and 1500 μmol mol^−1^, with 2–3 min between each step. The net rate of CO_2_ assimilation (A_N_), stomatal conductance (g_s_), the transpiration rate (E), and the sub-stomatal CO_2_ concentration (C_i_) were estimated at a PPFD of 1200 µmol m^−2^ s^−1^ and 400 µmol mol^−1^ [CO_2_] using equations developed in this study [54]. Estimations of the maximum carboxylation velocity of RuBisCO (V_cmax_) and the maximum electron transport rate contributing to RuBP regeneration (J_max_) were determined according to Sharkey et al. [45].

### 4.3. Nitrogen Labeling

Labeling was conducted during the period in which the plants were exposed to the different VPD treatments. For this purpose, a randomized subset of plants at each water and VPD level was ^15^N labeled by replacing KNO_3_ (1.22 g L^−1^) in the Hoagland solution with K_15_NO_3_ (same concentration). For each irrigation regime and VPD level, 4 pots were labeled during days 1, 2, and 4. During each labeling day, 10 mL (per pot) of the ^15^N-enriched Hoagland solution was provided. All the determinations were conducted in both labeled and unlabeled flag leaves collected during the last day of the labeling period (day 4).

### 4.4. Amino Acid Profile and Content

Lyophilized flag leaves (0.1 g DW) were ground with liquid N_2_ and homogenized with 1 mL of 1 M HCl. The extract was centrifuged at 20,000× *g* and 4 °C for 10 min. The supernatant was then adjusted to pH 7 (with NaOH). Amino acids were derivatized at room temperature for 15 h with fluorescein isothiocyanate (FITC) dissolved in 20 mM acetone/borate, pH 10. Single amino acids were determined by high-performance capillary electrophoresis using a Beckman Coulter PA-800 apparatus (Beckman Coulter, Inc., Brea, CA, USA) with laser-induced fluorescence detection (argon ion: 488 nm). Samples were derivatized with FITC and separation was performed in a 50 μm i.d. × 43/53.2 cm fused silica capillary at a voltage of 30 kV and a temperature of 20 °C. The migration buffer was 80 mM borax (pH 9.2) containing 45 mM alpha-cyclodextrin. Samples were injected using a pressurized method (5 s). Norvaline and glutamic acid was used as the internal standard. Identification of the peaks in the flag leaf samples was performed by comparing the migration time of each amino acid standard with the migration time of the detectable peaks in the flag leaf electropherogram.

### 4.5. N, C, and O Isotope Composition Analyses (δ ^15^N, δ ^13^C and δ ^18^O)

To determinate δ ^15^N, δ ^13^C, and δ ^18^O in the total organic matter (TOM), 1 mg of oven-dried (60 °C for 48 h) flag leaf material was used. To determine the δ ^13^C in the water-soluble compounds, 50 mg of dried flag leaf samples were suspended in 1 mL of distilled water in an Eppendorf tube (Eppendorf Scientific, Hamburg, Germany), mixed, and then centrifuged at 12,000× *g* for 5 min at 5 °C. After centrifugation, the supernatant containing the leaf water-soluble compounds was heated for 3 min at 100 °C and afterward the solution was put on ice for 3 min. The supernatant fraction was transferred to tin capsules for isotope analysis.

Nitrogen and C isotopic composition analyses were conducted using a flash 1112 Elemental Analyzer (Carbo Erba Instrumentazione, Milan, Italy) coupled to an IRMS Delta C isotope ratio mass spectrometer through a Conflo III Interface (Thermo-Finnigan, Bremen, Germany). The N and C isotopic compositions were determined as: [δ^XX^X = (R_sample_/R_standard_) − 1]. δ ^15^N and δ ^13^C accuracy were monitored using international secondary standards for ^15^N/^14^N (IAEA-N1 and IAEA-N_2_ ammonium sulfate and IAEA-NO3 potassium nitrate, IAEA, Wien, Austria) and ^13^C/^12^C (Vienna Pee Dee Belemnite calcium carbonate (V-PDB) ratios.

The leaf ^18^O/^16^O ratio was determined with an online pyrolysis technique using a thermo-chemical elemental analyzer (TC/EA Thermo Quest Finnigan, Bremen, Germany) coupled to an IRMS (Delta C Finnigan MAT, Bremen, Germany). Oxygen isotope composition was expressed with the same notation, δ ^18^O, as the N and C composition, with reference to IAEACH-6 (sucrose, d18OV-SMOW = 36.4‰).

### 4.6. Gene Expression

#### 4.6.1. RNA Isolation

Total RNA was isolated from 100 mg of flag leaves with a phenol/chloroform extraction method [55]. DNase treatment of total RNA and cDNA synthesis was performed according to Qiagen’s protocol (Quantitect Reverse Transcription KIT Cat#205311; Qiagen, CA, USA).

#### 4.6.2. Quantitative Real-Time Reverse Transcription Polymerase Chain Reaction (RT-qPCR)

The expression of the RuBisCO large subunit (RBCL), the RuBisCO activase β subunit (RcaB), the nitrate transporters NRT1.3 [17], NRT1.5, NRT1.8, NRT1.4, and NRT1.7 [19], NRT1.9 [18], glutamine synthetase (GS2), and NIR were studied with real-time PCR by using an iCycler (Bio-Rad, Hercules, CA, USA). cDNAs were obtained from 2.5 μg of total DNase-treated RNA in a 20 μL reaction containing oligo (dT)_15_ primer (Promega, Madison, WI, USA), 10 mM dNTP (Invitrogen, Carlsbad, CA, USA), 0.1 M DTT (Invitrogen, Carlsbad, CA, USA), 40 U of RNase inhibitor (Promega), 5× first strand buffer (Invitrogen), and 200 U of Superscript II Reverse Transcriptase (Invitrogen) with the temperature recommended by the enzyme supplier. The primer sets used to amplify each studied gene in the synthesized cDNAs, including that of reference gene actin, are shown in Appendix A.

Each 23 μL reaction contained 3 μL of a 1:10 dilution of the cDNA, 10.5 μL of Master Mix (Bio-Rad Laboratories S.A., Madrid, Spain), 8.6 μL of MilliQ H_2_O, and 0.45 μL of each primer pair. The PCR program consisted of a 3 min incubation at 95 °C to activate the hot-start recombinant Taq DNA polymerase, 31 cycles of 30 s at 94 °C, 30 s at the established annealing temperature, and 30 s at 72 °C, where the fluorescence signal was measured. The specificity of the PCR amplification procedure was checked with a heat dissociation protocol (from 70–100 °C) after the final PCR cycle.

Real-time PCR experiments were carried out with four biologically independent samples, with the threshold cycle (Ct) determined in triplicate. The relative levels of transcription were calculated by using the 2^−ΔΔCt^ method [56], using the actin gene as a reference with Ct variation among treatments being less than 1. All the primers used presented efficiencies close to 1.9 as described by Pfaffl [57]. Negative controls without cDNA were used in all PCRs.

### 4.7. Statistical Analysis

Gas exchange, amino acid profiles, gene expression, and isotopic composition analyses were carried out three times for each treatment (water regimes: WW/WS and high/low VPD). An analysis of variance (two-way ANOVA; two factors, two levels) was conducted in order to determine the effects of the treatments and their possible interactions. Differences among groups were tested with the Least Significant Differences (LSD) posthoc test. A significance threshold of 0.05 was used. In the case of gene expression data, since fold change values are not normally distributed, the 95% confidence intervals were used. Statistical analyses were performed using the SPSS statistical program. Appendix A show ANOVA P and F values of the different parameters analyzed in the Figures and in Table 1, respectively. Figures were created using the Sigma-Plot 11.0 program.

## 5. Conclusions

In summary, high atmospheric VPD has effects on durum wheat C and N metabolism even when plants have access to sufficient water in the substrate, and this impaired C metabolism and activated N metabolism. Water stress decreased stomatal conductance, photosynthetic electron transport, and RuBisCO large subunit and RuBisCO activase gene expression, changes that led to low photosynthetic rates, and that was accompanied by protein degradation. However, when the atmospheric VPD of water-stressed leaves was decreased, it ameliorated the effect of drought on C metabolism (increasing photosynthesis, related to increases in the photosynthetic electron transport rate), increased the N uptake in the mesophyll cells, and decreases the amount of N remobilized from the roots to the young leaves.

## Figures and Tables

**Figure 1 plants-10-00120-f001:**
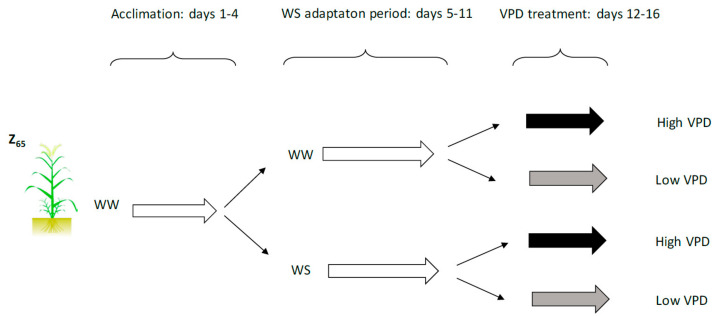
Experimental design on wheat plants exposed to different water irrigation (pot capacity-WW- *versus* 50% of pot capacity-WS) and ambient vapour pressure deficit (VPD; high 40 % RH *versus* low 80% RH). During acclimation (days 1–4) and WS adaptation (days 5–11) period plants were grown at 25/18 °C RH and 1.27/0.42 Kpa VPD (corresponding to 60/80 % RH). During days 12–16 both control plants and water stressed plants were exposed to high (2.27/0.62 Kpa; 40/70 % RH) and low (1.01/0.36 Kpa; 70/80 % RH) VPD levels. During this final period 15N labelling was carried out in all treatments.

**Figure 2 plants-10-00120-f002:**
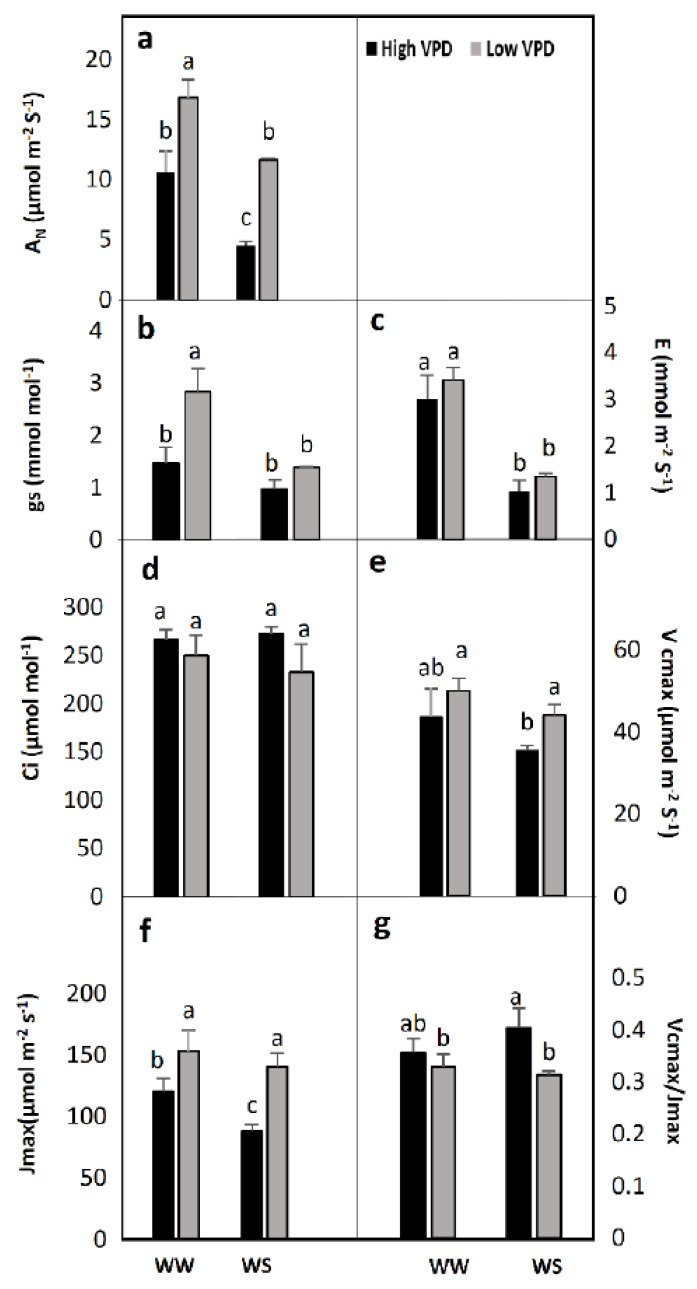
Effect of water irrigation (pot capacity-WW- *versus* 50% of pot capacity-WS) and ambient vapour pressure deficit (VPD; high 40 % RH *versus* low 80% RH) on (**a**) net photosynthesis (AN), (**b**) leaf conductance (gs), (**c**) transpiration (E), (**d**) intercellular CO_2_ concentration (Ci), (**e**), Rubisco maximum carboxylation capacity (Vcmax) (**f**) maximum rate of electron transport (Jmax) and (g) the Vcmax/ Jmax rate (**g**) determined on wheat flag leaves. Each value represents the mean ± se of four replications. Statistical analysis was made by a two-factor ANOVA. Different letters above the columns indicate significant differences (*p* < 0.05) between treatments as determined by LSD test.

**Figure 3 plants-10-00120-f003:**
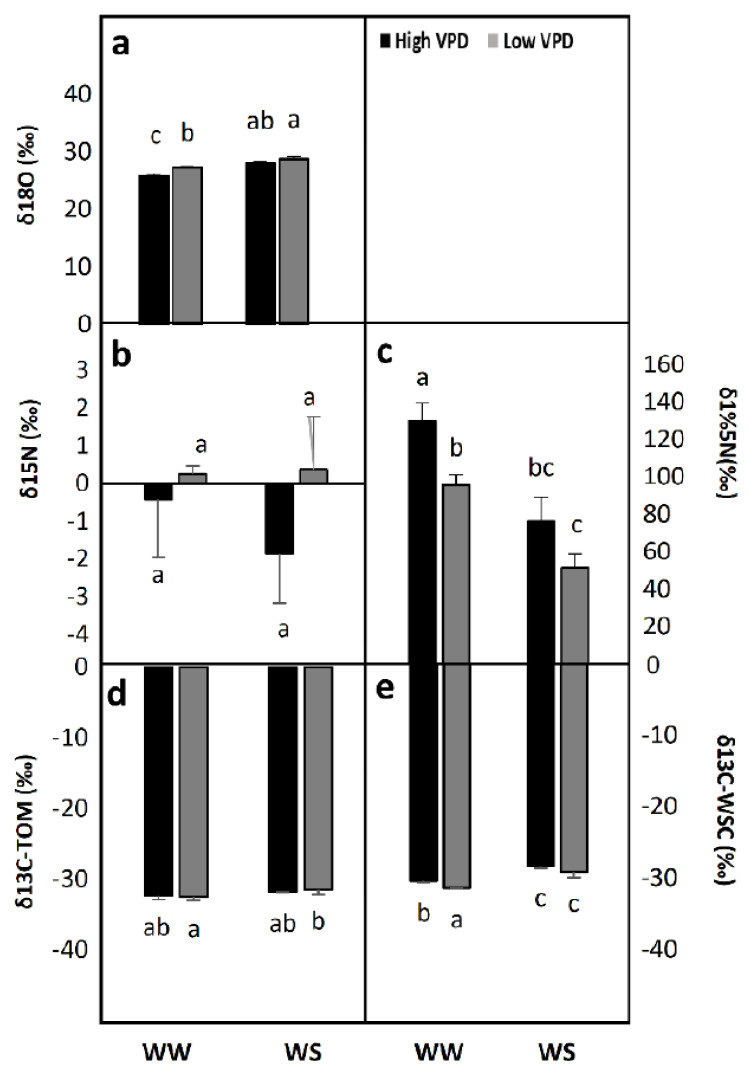
Effect of water irrigation (pot capacity-WW- *versus* 50% of pot capacity-WS) and ambient vapour pressure deficit (VPD; high 40 % RH *versus* low 80% RH) on (**a**) 18O isotopic composition (δ18O), 15N isotopic composition (δ15N) of (**b**) non labelled and (**c**) 15N labelled plants, 13C isotopic composition (δ13C) in (**d**) total organic matter (TOM) and (**e**) in water soluble content (WSC) determined on wheat flag leaves. Each value represents the mean ± se of four replications. Statistical analysis was made by a two-factor ANOVA. Different letters above the columns indicate significant differences (*p* < 0.05) between treatments as determined by LSD test.

**Figure 4 plants-10-00120-f004:**
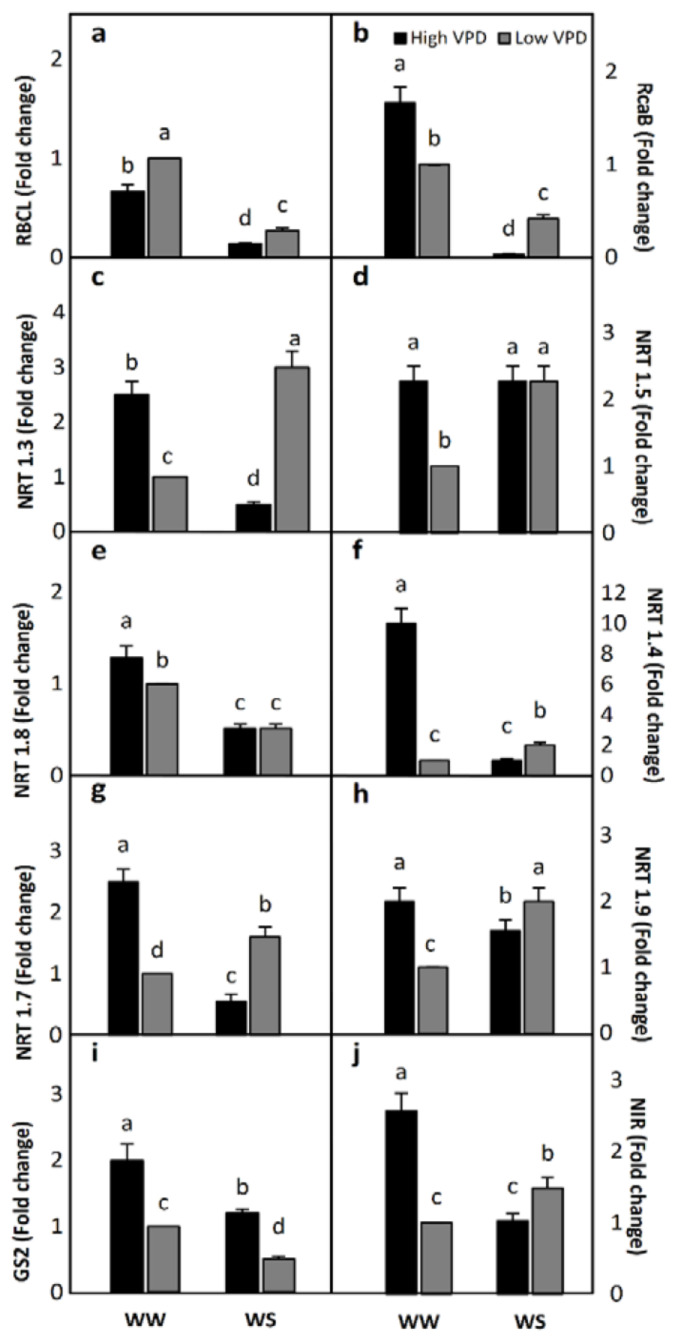
Effect of water irrigation (pot capacity-WW- versus 50% of pot capacity-WS) and ambient vapour pressure deficit (VPD; high versus low) on gene expression of RuBisCO (**a**) Rubisco large subunit (RBCL), (**b**) Rubisco activase β subunit (RcaB), nitrate transporters (**c**) NRT1.3, (**d**) NRT1.5, (**e**) NRT1.8, (**f**) NRT1.4, (**g**) NRT1.7, (**h**) NRT1.9, (**i**) glutamine synthetase (GS2) and (**j**) nitrite reductase (NIR) determined on wheat flag leaves. Each value represents the fold change with respect to well-watered low VPD plants. Four replications were used. Different letters above the plots indicate significant differences. Since fold change values are not normally distributed, the 95% confidence intervals are shown.

**Figure 5 plants-10-00120-f005:**
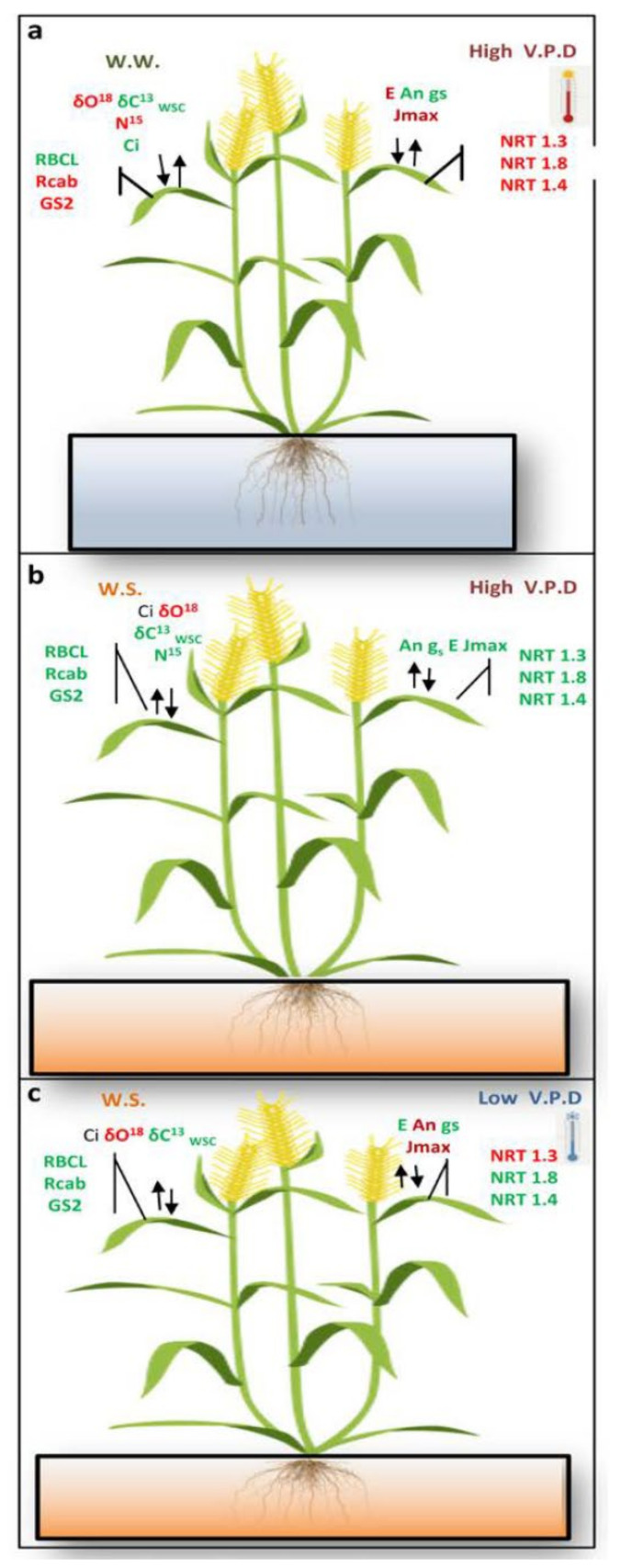
Model for whole plant C and N metabolism in durum wheat plants grown under low/high atmospheric VPDs and water availability: (**a**) well watered plants grown under high atmospheric VPD, (**b**) plants grown in drought conditions and high atmospheric VPD, (**c**) plants under water stressed conditions and low atmospheric VPD. Colors indicate either decreases (green) or increases (red).

**Table 1 plants-10-00120-t001:** Effect of water irrigation (pot capacity-WW- *versus* 50% of pot capacity-WS) and ambient vapour pressure deficit (VPD; high 40 % RH *versus* low 80% RH) on wheat leaf nitrogen (N, %) and amino acid (μg g^−1^ FW) content: arginine (ARG), lysine (LYS), Leucine (LEU), isoleucine (ILE), methionine (MET), Phenylalanine (PHE), Tryptophan (TRP), histidine (HIS), Tyrosine (TYR), Valine (VAL), glutamine (GLN), proline (PRO), asparagine (ASN), GABA, threonine (THR), serine (SER), glycine (GLY), alanine (ALA), glutamic acid (GLU), aspartic acid (ASP) determined on wheat flag leaves. Each value represents the mean ± se of four replications. Statistical analysis was made by a two-factor ANOVA. Different letters above the columns indicate significant differences (*p* < 0.05) between treatments as determined by LSD test. N content and amino acids that, compared with WW Low VPD, were significantly altered by irrigation and/or relative humidity treatments are highlighted in orange colour, whereas the amino acids that were not significantly affected are highlighted in blue colour.

Parameter	WW High VPD	WW Low VPD	WS High VPD	WS Low VPD
**N%**	2.85 ± 0.31 **a**	2.9 ± 0.37 **a**	2.91 ± 0.31 **a**	2.73 ± 0.25 **a**
**ARG**	1134.33 ± 122.4 **b**	581.3 ± 23.7 **c**	2346.7 ± 265.1 **a**	2041.7 ± 192.7 **a**
**LYS**	386 ± 91.6 **b**	106.3 ± 14.3 **c**	854.7 ± 86.3 **a**	621.7 ± 94.3 **ab**
**LEU**	186.6 ± 33.3 **b**	55 ± 12.4 **c**	269 ± 32.8 **ab**	356.7 ± 45.6 **a**
**ILE**	199.66 ± 47.2 **b**	57.3 ± 9.8 **c**	365.7 ± 34 **a**	410.7 ± 56.8 **a**
**MET**	46 ± 3.5 **bc**	7.67 ± 1.3 **c**	164.3 ± 62.7 **b**	310.7 ± 50.4 **a**
**PHE**	578.3 ± 59.4 **b**	186.3 ± 45.7 **c**	1456.7 ± 347.2 **a**	1453 ± 113.2 **a**
**TRP**	619 ± 38.9 **b**	104.6 ± 43.7 **c**	1375 ± 320.4 **a**	1101 ± 173.5 **a**
**HIS**	155.6 ± 31.3 **bc**	73.3 ± 12.4 **c**	275 ± 14.7 **a**	194 ± 31.2 **ab**
**TYR**	202 ± 1.7 **b**	74.3 ± 9.8 **c**	520.3 ± 49.8 **a**	479.3 ± 77.8 **a**
**VAL**	530 ± 162.5 **b**	124 ± 25.7 **c**	784 ± 101.8 **ab**	1099.3 ± 153.5 **a**
**GLN**	4982.6 ± 885.7 **b**	1406.3 ± 225.8 **c**	6460.7 ± 476.9 **b**	10829.5 ± 967.5 **a**
**PRO**	1413.6 ± 148.8 **b**	158.3 ± 24.9 **c**	5515 ± 1801.7 **a**	5507 ± 1334.3 **a**
**ASN**	160.5 ± 6.5 **b**	234.6 ± 49.5 **b**	781.3 ± 91.3 **a**	388.3 ± 62.4 **b**
**GABA**	204.3 ± 67.4 **a**	219.6 ± 59 **a**	255 ± 70 **a**	284.3 ± 59.6 **a**
**THR**	250 ± 19.9 **b**	176 ± 23 **b**	286 ± 28.3 **a**	319 ± 48.4 **a**
**SER**	1446.5 ± 12.5 **b**	789 ± 154 **c**	1939.33 ± 63.8 **a**	1371.3 ± 178.3 **b**
**GLY**	127 ± 11 **b**	100.6 ± 19.5 **b**	142 ± 20.1 **ab**	206.7 ± 37.3 **a**
**ALA**	1366.3 ± 214.9 **a**	1604.6 ± 140.8 **a**	1424.3 ± 198.3 **a**	713.5 ± 5.5 **b**
**GLU**	2999.6 ± 65.8 **a**	2476 ± 96 **b**	2636.5 ± 1.5 **ab**	2599.7 ± 178.5 **b**
**ASP**	1546 ± 160.6 **a**	776 ± 7.3 **b**	1267 ± 138.6 **ab**	1106.3 ± 172.8 **ab**

## Data Availability

Data is contained within the article or Appendix A.

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
