# Peer review of "Short-Term Exposure to High Atmospheric Vapor Pressure Deficit (VPD) Severely Impacts Durum Wheat Carbon and Nitrogen Metabolism in the Absence of Edaphic Water Stress"

_plants, 2021, doi:10.3390/plants10010120_

Round 1

Reviewer 1 Report

Major changes

  1. There are major issues with the wording of the results being inaccurate/misleading/unclear. The authors need to carefully review their descriptions of their results before this manuscript is suitable for publication e.g.
    • L98 WW high VPD has similar An to WS low VPD
    • L100 it is apparent from Fig 2b that water stressed plants don’t show a significant decrease in gs with increased VPD as they show relatively low gs under both VPD conditions
    • L119 +120 Shouldn’t this read Fig.3a?
    • L146 shouldn’t unlabelled be Fig 3b?
    • L148 in water stressed plants, low under both VPD conditions
    • L147 Fig 3c
    • L150 Fig 3b
    • L154 No significant differences were observed in response to increases in VPD under well-watered conditions or water stressed conditions
    • L159 this sentence does not match the statistical groupings shown on Fig. 4b
    • L193 [appear to be] the phases most affected by water deficiency and aerial VPD level [based on these gene expression results]
    • L199 only sig difference is between WW and WS under high VPD
    • L208 Table 1 indicates THR and GLY are affected by water availability and that GLU under WW was affected by VPD
    • L212 water stress [and/or] VPD (not all affected by both).
  2. The reason that the specific Triticum genes examined in this study were chosen needs to be clarified and justified. Have the proteins encoded by these specific Triticum genes been shown to have the roles discussed (in which case these studies need to be cited)? Or have these genes been selected based on orthology to genes characterised in other species (e.g. Arabidopsis) – in which case this needs to be clearly stated (with relevant studies cited) and orthology clearly presented (with phylogenetic relationships shown; or studies that show phylogenetic relationships cited). Where there are multiple co-orthologs in durum wheat, the choice to measure one of these and not the others needs to be justified, or all should be measured. The Figure 4 legend/axis needs to include that the figure is showing gene expression and exactly what values are displayed specifying the units/normalisation. Gene names should be italicised.
  3. The authors should include their p-values for all statistical tests (e.g. in supp tables) and double check their labelling of significant difference categories particularly in Fig2d+e and Fig3a+c as some appear to be mislabelled.

Minor changes:

  1. Some small typos/grammatical errors throughout, including (suggested change in square brackets):
    • L39: [On] the other hand
    • L54: [stomatal] cavities
    • L55: [a] proxy
    • L59: VPD [drives] global
    • L85: genotypes [tend] …etc
  2. Figures need to be renumbered and figure parts reorganised to match in-text presentation order.
  3. It would make more sense to display 80% RH on the left, and 40% (high VPD) on the right to match with WW vs WS order.
  4. Higher dpi is needed for figures – these currently don’t display optimally.
  5. L95 + 105 + where relevant throughout: As plants weren’t moved from high to low VPD, the authors need to reword to clarify that these traits had e.g. higher values under low VPD than under high VPD (rather than saying they improved/increased under low VPD). (Unless I’ve misunderstood the author’s methods, in which case these should be clarified.)
  6. L121 sentence meaning is unclear
  7. Please make gene names correct + consistent throughout, particularly for NRT1.5, NRT1.8 and NRT1.9 (sometimes with extra letter, sometimes not)
  8. Some figure parts are not mentioned in text – if the authors don’t consider these important enough to mention maybe these should be moved to the supp data?
  9. Fig3b the y-axis is mislabelled
  10. Table 1: it would help if the cells were colour-coded according to statistical grouping so that differences can be seen more easily. WW40 is misaligned.
  11. L239-L244 this would be better at the start of the results or in the introduction.
  12. Wording L400 should be clarified to state that the “actin gene was used as reference [as it was found to show] Ct variation of less than 1 between treatments” – currently the wording is confusing.
  13. Figure 1 – please include VPD details on acclimation days and WS adaptation period for clarity.
  14. L328 Fig1 says RH 80% - please clarify.

Author Response

Major changes

  1. There are major issues with the wording of the results being inaccurate/ misleading/unclear. The authors need to carefully review their descriptions of their results before this manuscript is suitable for publication e.g.

Following the recommendation made by the referee. The manuscript wording has been revised. Further, in order to improve the understanding of the paper, it has been revised by an Australian manuscript Editing company (called Out of Site).

L98 WW high VPD has similar An to WS low VPD

The sentence has been revised accordingly.

L100 it is apparent from Fig 2b that water stressed plants don’t show a significant decrease in gs with increased VPD as they show relatively low gs under both VPD conditions

The text has been modified in agreement with the comment made.

L119 +120 Shouldn’t this read Fig.3a?

Correct. The reference to Figure 3 has been updated.

L146 shouldn’t unlabelled be Fig 3b?

Correct. The reference to Figure 3 has been updated.

L148 in water stressed plants, low under both VPD conditions

The sentence has been modified accordingly.

L147 Fig 3c

Modified.

L150 Fig 3b

Modified.

L154 No significant differences were observed in response to increases in VPD under well-watered conditions or water stressed conditions.

The sentence has been revised.

L159 this sentence does not match the statistical groupings shown on Fig. 4b.

Following the recommendation made, the text has been revised in order to properly match the results obtained

L193 [appear to be] the phases most affected by water deficiency and aerial VPD level [based on these gene expression results]

Authors agree with the observation and the text has been revised accordingly.

L199 only sig difference is between WW and WS under high VPD

The text has been modified following the comment made.

L208 Table 1 indicates THR and GLY are affected by water availability and that GLU under WW was affected by VPD

The text has been revised.

L212 water stress [and/or] VPD (not all affected by both).

The text has been revised.

  1. The reason that the specific Triticum genes examined in this study were chosen needs to be clarified and justified. Have the proteins encoded by these specific Triticum genes been shown to have the roles discussed (in which case these studies need to be cited)? Or have these genes been selected based on orthology to genes characterised in other species (e.g. Arabidopsis) – in which case this needs to be clearly stated (with relevant studies cited) and orthology clearly presented (with phylogenetic relationships shown; or studies that show phylogenetic relationships cited). Where there are multiple co-orthologs in durum wheat, the choice to measure one of these and not the others needs to be justified, or all should be measured. The Figure 4 legend/axis needs to include that the figure is showing gene expression and exactly what values are displayed specifying the units/normalisation.

In the current study, we selected the NRT1 transporter family because they are low affinity transporters (Buchner and Hawkesford 2014) and in our experimental conditions no nitrogen limitation was present. Also Guo et al. (2019) found that NRT1 genes had more relevance regarding wheat growth than NRT2 ones. Also, the genes selected are the most expressed in shoot tissues of wheat plants (Buchner and Hawkesford 2014).

Buchner P, Hawkesford MJ (2014) Complex phylogeny and gene expression patterns of members of the NITRATE TRANSPORTER 1/PEPETIDE TRANSPORTER family (NPF) in wheat. J Exp Bot 65: 5697-5710.

Guo H, Tian Z, Sun S, Li Y, Jiang D, Cao W, Dai T (2019) Preanthesis root growth and nitrogen iptake improved wheat grain yield and nitrogen use efficiency. Agron J 111: 3048-3056.

Gene names should be italicised.

The legend axis of figure 4 were corrected.

  1. The authors should include their p-values for all statistical tests (e.g. in supp tables) and double check their labelling of significant difference categories particularly in Fig2d+e and Fig3a+c as some appear to be mislabelled.

As requested, statistic analyses labelling has been revised and P/f-values for all parameters have been included in supplementary tables (Table 2S and 3S).

Minor changes:

  1. Some small typos/grammatical errors throughout, including (suggested change in square brackets):

L39: [On] the other hand

Revised.

L54: [stomatal] cavities

Revised.

L55: [a] proxy

Revised.

L59: VPD [drives] global

Revised.

L85: genotypes [tend] …etc

Revised.

  1. Figures need to be renumbered and figure parts reorganised to match in-text presentation order.

The figure numbers have been revised accordingly.

  1. It would make more sense to display 80% RH on the left, and 40% (high VPD) on the right to match with WW vs WS order.

While we might agree with suggestion made by the referee, we also think that keeping the current version contributes to more clearly identify potential differences on the different parameters.

  1. Higher dpi is needed for figures – these currently don’t display optimally.

The figures have been revised in order to improve their quality.

  1. L95 + 105 + where relevant throughout: As plants weren’t moved from high to low VPD, the authors need to reword to clarify that these traits had e.g. higher values under low VPD than under high VPD (rather than saying they improved/increased under low VPD). (Unless I’ve misunderstood the author’s methods, in which case these should be clarified.)

The text has been modified in order to make it more understandable.

  1. L121 sentence meaning is unclear.

The sentence has been revised.

  1. Please make gene names correct + consistent throughout, particularly for NRT1.5, NRT1.8 and NRT1.9 (sometimes with extra letter, sometimes not).

The extra letter was removed.

  1. Some figure parts are not mentioned in text – if the authors don’t consider these important enough to mention maybe these should be moved to the supp data?

After some discussion on this point, the authors think that while no target information is reflected in some figures sections, keeping the data together might provide a more integrated vision to readers. For this reason, we consider that it could be interesting to keep the figures in the current form.

  1. Fig3b the y-axis is mislabelled.

Figure has been revised accordingly.

  1. Table 1: it would help if the cells were colour-coded according to statistical grouping so that differences can be seen more easily. WW40 is misaligned.

As requested, Table 1 has been revised coloured. Also, data on WW 40 treatment has been aligned.

  1. L239-L244 this would be better at the start of the results or in the introduction.

While we agree with the potential interest on moving this section, we still think that the current version contributes to explain how the different treatments etc. were applied.

  1. Wording L400 should be clarified to state that the “actin gene was used as reference [as it was found to show] Ct variation of less than 1 between treatments” – currently the wording is confusing.

The manuscript has been revised.

  1. Figure 1 – please include VPD details on acclimation days and WS adaptation period for clarity.

As requested, VPD data during acclimation and adaptation has been included in Figure 1.

  1. L328 Fig1 says RH 80% - please clarify.

The text has been revised.

Reviewer 2 Report

The manuscript "Short exposure to high atmospheric vapor pressure deficit (VPD) severely impacts durum wheat carbon and nitrogen metabolism in absence of edaphic water stress" by Fakhet et al. submitted for publication in Plants is focused on durum wheat, VPD and its impact on carbon and nitrogen metabolism. Alongside physiological and transcriptional measurements have been performed. The manuscript is well written and will be of importance for the scientific community working on the topic. The work deserves to be published after revision and after addressing the comments and questions below.

1) Introduction

The introduction is well written and target the main problems but I would suggest to the authors to include a few words about the NRT.

2) Results

  • Figure 5 needs to be in higher quality because is not really readable.
  • Amino acid abbreviations should be the same each time (example Line 210: sometimes is ARG sometimes Lys etc.
  • I think the most correct way to write is: RuBisCO although Rubisco is also widely used in the scientific community.
  • NRT transporters is  a large family with NRT1, 2, 3 and divided into high and low affinity nitrate transporters. And this should be mentioned in the text. The authors studied the expression of only NRT1, why didn't they included some others (for example NRT2)?
  • Fig.4 Is the expession level of the studied genes presented as relative expression or fold change? Delta-delta Ct presents fold change. This has to be clarified in the text and on the figure 4.
  • Table 1: the caption is heavy to read, no need to repeat for each AA mg g-1. And what is N on the first row total nitrogen or..?This needs clarification.
  • L208-212: is it necessary to write the full names of all amino acids?
  • I wonder if the authors have been thinking to measure the enzymatic activity of nitrate reductase?

3) Materials & Methods

MM are well described but I have some comments.

  • I think the full composition of Hoegland solution should be mentioned. Did the authors used NH4 or only NO3 as N source?
  • Line 324: it's written "tow RH". Is that correct or should be low?
  • The number of plants used in the experiment should be given.
  • How many mg of froezn flag leaf material was used for amino acid profiling and total RNA extraction?
  • Gene names should be in italic.
  • Line 381: it should be written RT-qPCR instead of RT-PCR
  • RT-qPCR amplification conditions described between L391-L397 are not really clear and need to be rewritten in more comprehensive way.
  • Did the authors really used 31 cycles for the amplification?

4) Conclusions

The conclusion supports the results.

5) Reference list

The reference list needs to be unified (for example: sometimes the names are in italic sometimes not, also reference 55, the journal name is Methods not Method)

Author Response

The manuscript "Short exposure to high atmospheric vapour pressure deficit (VPD) severely impacts durum wheat carbon and nitrogen metabolism in absence of edaphic water stress" by Fakhet et al. submitted for publication in Plants is focused on durum wheat, VPD and its impact on carbon and nitrogen metabolism. Alongside physiological and transcriptional measurements have been performed. The manuscript is well written and will be of importance for the scientific community working on the topic. The work deserves to be published after revision and after addressing the comments and questions below.

The authors would like to thank the revision and evaluation made on our manuscript.

1) Introduction

- The introduction is well written and target the main problems but I would suggest to the authors to include a few words about the NRT.

In agreement with the comment made, an introduction to NRTs has been included in the Introduction section.

2) Results

- Figure 5 needs to be in higher quality because is not really readable.

The quality of figures has been improved.

- Amino acid abbreviations should be the same each time (example Line 210: sometimes is ARG sometimes Lys etc.

Amino acid abbreviations have been revised.

- I think the most correct way to write is: RuBisCO although Rubisco is also widely used in the scientific community.

The reference to RuBisCO has been revised throughout the text.

- NRT transporters is a large family with NRT1, 2, 3 and divided into high and low affinity nitrate transporters. And this should be mentioned in the text. The authors studied the expression of only NRT1, why didn't they included some others (for example NRT2)?

As explained before, in the current study, we selected the NRT1 transporter family because they are low affinity transporters (Buchner and Hawkesford 2014) and in our experimental conditions no nitrogen limitation was present. Also Guo et al. (2019) found that NRT1 genes had more relevance regarding wheat growth than NRT2 ones. Also, the genes selected are the most expressed in shoot tissues of wheat plants (Buchner and Hawkesford 2014).

Buchner P, Hawkesford MJ (2014) Complex phylogeny and gene expression patterns of members of the NITRATE TRANSPORTER 1/PEPETIDE TRANSPORTER family (NPF) in wheat. J Exp Bot 65: 5697-5710.

Guo H, Tian Z, Sun S, Li Y, Jiang D, Cao W, Dai T (2019) Preanthesis root growth and nitrogen iptake improved wheat grain yield and nitrogen use efficiency. Agron J 111: 3048-3056.

- Fig.4 Is the expession level of the studied genes presented as relative expression or fold change? Delta-delta Ct presents fold change. This has to be clarified in the text and on the figure 4.

It is fold change and it has been modified in the new version of the manuscript.

- Table 1: the caption is heavy to read, no need to repeat for each AA mg g-1. And what is N on the first row total nitrogen or..? This needs clarification.

Table caption has been revised in order to make it more understandable.

- L208-212: is it necessary to write the full names of all amino acids?

The text has been modified accordingly.

- I wonder if the authors have been thinking to measure the enzymatic activity of nitrate

reductase?

While we agree with the fact that analysing NR activity could have been of interest for the study, unfortunately, this analyses was not taken into account. However, we will keep it in mind for future experiments.

3) Materials & Methods

MM are well described but I have some comments.

- I think the full composition of Hoagland solution should be mentioned. Did the authors used NH4 or only NO3 as N source?

Yes, as it is described in the new version NO3 was used as N source.

- Line 324: it's written "tow RH". Is that correct or should be low?

The text has been revised accordingly.

- The number of plants used in the experiment should be given.

As it has been specified in the new version, 3 plants were used in the experiment.

- How many mg of froezn flag leaf material was used for amino acid profiling and total RNA extraction?

As it is noted in the new version, in both analyses, 100 mg were used.

- Gene names should be in italic.

The text has been revised.

- Line 381: it should be written RT-qPCR instead of RT-PCR.

Revised.

- RT-qPCR amplification conditions described between L391-L397 are not really clear and need to be rewritten in more comprehensive way.

The manuscript has been revised in order to make it more understandable.

- Did the authors really used 31 cycles for the amplification?

Yes.

4) Conclusions

The conclusion supports the results.

5) Reference list

The reference list needs to be unified (for example: sometimes the names are in italic sometimes not, also reference 55, the journal name is Methods not Method).

References have been revised and unified.

Reviewer 3 Report

The work of Fakhet and coworkers asks relevant questions and presents interesting data. However, accessibility and readability are greatly impaired by a lack of proper presentation. I cannot help the feeling that the paper being finished too hasty and without a careful pre-submission check for errors.   Unfortunately, this makes a potentially interesting manuscript annoying to read. At this stage I, therefore suggest to focus in a first revision on improving the results section and to present the data in a more comprehensible way. This would help - at least this reviewer - to rather engage in the data than in the form. Below are a number of points that might help the authors in revising their manuscript.

General remarks:

  • The resolution of Figures and of axis labeling is too low to read. In particular Figure 5 is not readable at all. I could therefore only review the parts I could read. Please increase resolution of Figures.
  • I have serious problems with the results section. It read completely different from the other sections and is extremely hard to follow through. It would massively help to provide a short introduction sentence in front of each results paragraph. What exactly has been measured here? Why has it been measured? Which questions will be answered?
  • Please introduce abbreviations upon first use (e.g. TOM)

Specific comments:

Please go through the entire manuscript and correct syntax and spelling errors. Here are a few examples (not a complete list):

  • L 48: check syntax and rephrase
  • L 57: coworker(s) ?
  • L59: dive?
  • L80: wheat plant(s) mature
  • L81: “being it remarkable” change to “notably in the flag leaf”
  • L 95-106: whole paragraph; in particular change “cause” to “caused”, “no” to “not”, “tent” to “tended”,
  • ….
  •  

Table 1: The units for the amino acids (mg/g) are apparently wrong (higher than 1000 for some). Please correct. In addition, I suggest displaying amino acids in form of a heat map.

Figure 1 really needs to be the first figure (the experimental setup), along with a short description of the whole experiment. Explain abbreviations in the legend (WS, WW)

Authors describe Figure 3d and 3e in results before 3a and 3b in 2.2.2 and 2.2.3, respectively. Please check order.

Figure 4: NRT1.3 expression decreased under well-watered conditions with low VPD, however, N-labeling increased. Could you maybe explain this result? (same question for L296)

  • L159: Authors write RcAB expression “significantly increased”, however, no significant difference is visible from the statistical analysis given in corresponding Figure 4B. Please check the statistics again.

  • L267: data shows the difference between well-watered and water stress but does not show significant differences between different VPDs.

Methods 4.4., Amino acids:

The pH was “fixed”, change to “adjusted to”. Also “7-9” seems to be a very broad range to me. Also, please describe the measurement method in a more detailed way or cite corresponding literature. Which standards were used to determine the amounts of the individual amino acids?

Author Response

Reviewer 3

General remarks:

  • The resolution of Figures and of axis labeling is too low to read. In particular Figure 5 is not readable at all. I could therefore only review the parts I could read. Please increase resolution of Figures.

The resolution of figures has been increased.

  • I have serious problems with the results section. It read completely different from the other sections and is extremely hard to follow through. It would massively help to provide a short introduction sentence in front of each results paragraph. What exactly has been measured here? Why has it been measured? Which questions will be answered?

While the Results has been revised, we consider that the current version could be understandable for readers. Adding introduction sections to each paragraph could affect the flow of text. 

  • Please introduce abbreviations upon first use (e.g. TOM).

As requested, when required, abbreviations have been added to the text.

Specific comments:

Please go through the entire manuscript and correct syntax and spelling errors.

The manuscript has been revised by ourselves and by an Australian English Editing Company.

Table 1: The units for the amino acids (mg/g) are apparently wrong (higher than 1000 for some). Please correct. In addition, I suggest displaying amino acids in form of a heat map.

While we agree with the fact that it could be interesting to modify units for the most abundant amino acids, in order to reduce the length of Table 1 legend, we think that keeping the same units for all amino acids could be helpful for readers. Amino acids have been coloured in order to distinguish significant and non significant ones.

Figure 1 really needs to be the first figure (the experimental setup), along with a short description of the whole experiment. Explain abbreviations in the legend (WS, WW).

Figure 1 has been revised accordingly.

Authors describe Figure 3d and 3e in results before 3a and 3b in 2.2.2 and 2.2.3, respectively. Please check order.

In agreement with the comment made, Figure 3 legends and Results have been revised.

Figure 4: NRT1.3 expression decreased under well-watered conditions with low VPD, however, N-labeling increased. Could you maybe explain this result? (same question for L296)

The text has been revised following the recommendation made by the referee.

  • L159: Authors write RcAB expression “significantly increased”, however, no significant difference is visible from the statistical analysis given in corresponding Figure 4B. Please check the statistics again.

The text has been revised.

  • L267: data shows the difference between well-watered and water stress but does not show significant differences between different VPDs.

The text has been revised.

Methods 4.4., Amino acids:

The pH was “fixed”, change to “adjusted to”. Also “7-9” seems to be a very broad range to me. Also, please describe the measurement method in a more detailed way or cite corresponding literature. Which standards were used to determine the amounts of the individual amino acids?

The methodology used to determine amino acids has been revised.

Reviewer 4 Report

This revised manuscript is relatively easy to follow, and although I have some minor comments and criticisms, I recommend that the manuscript be accepted. I do think that the relevance of the study could be better emphasized, though this is perhaps the nature of such a descriptive study and I do not think it is necessary to resolve.

I have two minor suggestions that could be addressed for publication (but which I do not believe require an additional round of reviews).

  1. Fig 1 nicely outlines the experimental design, but the VPD treatments at right are too small. Increasing the font size would suffice but the VPD treatments could perhaps be better labeled with descriptions for high and low VPD under the VPD treatment heading, as is done for the watering regimes. Unique identifiers could then be given to each of the four treatment (e.g. WW, hVPD).
  2. The table 1 colors do not reflect the description; blue appears to indicate significant differences. The meaning of the significant differences is also unclear. It appears that the authors have chosen the treatment with the highest value as the comparison for significant difference, which is different for each amino acid. Instead, it would be easier to interpret if the comparison were to the same treatment, perhaps to the well-watered, low VPD treatment.

For future studies, the authors might consider describing multi-factor treatments by each factor of the ANOVA, e.g. whether the watering treatment, VPD treatment, and their interactions are significant. The description of the experiments here, by comparing each treatment interaction, makes it difficult to interpret the broader meaning of the data.

I am a bit concerned that for the qPCR data, the authors note that the fold change values are not normally distributed. I appreciate the transparency of the statement, but if the data were not normally distributed, this violates an assumption of the ANOVA test used to assess significance. An alternative could be to report qPCR expression levels relative to the reference/housekeeping gene (e.g., actin) and to perform statistics on these relative levels, which might resolve issues with the distribution of fold change values.

Author Response

I have two minor suggestions that could be addressed for publication (but which I do not believe require an additional round of reviews).

  • The authors would like to thank the comment made by the referee.

Fig 1 nicely outlines the experimental design, but the VPD treatments at right are too small. Increasing the font size would suffice but the VPD treatments could perhaps be better labeled with descriptions for high and low VPD under the VPD treatment heading, as is done for the watering regimes. Unique identifiers could then be given to each of the four treatment (e.g. WW, hVPD).

  • In agreement with the comment made, VPD labelling has been removed from Figure 1 and the corresponding details have been added to the Figure legend.

The table 1 colors do not reflect the description; blue appears to indicate significant differences. The meaning of the significant differences is also unclear. It appears that the authors have chosen the treatment with the highest value as the comparison for significant difference, which is different for each amino acid. Instead, it would be easier to interpret if the comparison were to the same treatment, perhaps to the well-watered, low VPD treatment.

  • As requested by the Referee, compared with WW-Low VPD treatment, treatments that did not significantly differ were highlighted in blue whereas the ones that were significanlty affected were highlighted in orange.

Round 2

Reviewer 1 Report

The authors have improved the English, but unfortunately many issues with the presentation of scientific content remain, including some I previously raised. Rather than writing "the text has been revised" the authors should say where/how they revised the text to address the concer, because there are too many instances where the authors claimed to have made the change, but did not in fact make the change. The authors need to be proactive in checking that their text is consistent with the data they are presenting. Considerable changes are required before this manuscript would be acceptable for publication in any journal. The recommendations below are in order of the appearance in the text, rather than importance.

  1. Figure citations do not match figure numbers (e.g. Fig1a is actually referring to Fig2a) – this needs to be fixed throughout. The order of panels in figures should also match the order discussed in text.
  2. Please add an in-text citation to Figure 1 (experimental) with a brief (1 or 2 sentence) explanation of experimental design at the start of the results. The explanation in the discussion could be moved up – as it would help readers to have this information before reading the results, rather than after.
  3. It would help if the temperature regime was also shown on Figure 1.
  4. I stand by my previous comment that it would be more intuitive to have high VPD (40% RH) displayed on the right of low VPD (80%), but if the authors wish to retain this order, they should label the legend as VPD high/low instead of RH%, to match how they describe these groups in the text.
  5. Figures are still appearing in pdf form at low resolution – which is particularly a problem for Figure 4 and Figure 5 where gene names are difficult to read or illegible, respectively. Increasing text size within figures may be sufficient to fix this issue if dpi cannot be further increased.
  6. L98 Higher AN values were found in low VPD conditions compared to high VPD conditions – the way that the sentence is currently worded implies that WS + low VPD was equal or second to WW + low VPD, whereas in reality it is ~ equal to WW + high VPD.
  7. L103 The authors still have not mentioned that gs likely did not decrease significantly with increased VPD in WS plants because stomata were already more closed than in WW.
  8. The authors are still referring to low VPD as “increasing” (e.g. L106, L149) or “decreasing” (L217) factors while Figure 1 shows that low VPD is a continuation of the VPD conditions during the adaptation period – this is thus similar to a control condition. The authors should instead either refer to the high VPD condition as “increasing” or “decreasing” factors, or the low VPD condition as being increased/decreased relative to the high VPD condition.
  9. Figure 3 label – x axis labels are not aligned.
  10. Significance values still appear to be mislabelled in Figure 3a (WS high VPD is shown as ab but is closer in value to c than to either a or b), Fig3c and Fig2d and Fig 2e. The authors should provide the posthoc test results to support their labelling.
  11. Figure 3 significance labels are not correctly aligned.
  12. The authors still need to cite the previous studies that isolated/characterised the function of the wheat genes included in qPCR experiments in this study in the introduction/results (e.g. L156, L177) with an introduction to the wheat gene families justifying their choice of study genes. Neither of the papers that they cited in the response to reviewers mentions NRT1.3-NRT1.9. If the names of these genes have been changed, both names should be provided at least in the supplementary material.
  13. L158 Fig3b no longer shows a significant difference in response to increased VPD under water stress conditions, according to the author’s updated significance labels.
  14. L181 NRT1.5A – the authors need to be consistent with their usage of gene names.
  15. The authors need to be careful of their descriptions of their gene expression results – they are presenting data for gene expression levels - genes encode proteins not functions, and the predicted effect of changes in gene expression on processes is an extrapolation and this needs to be clear.
    1. L182 a gene cannot “code for nitrate loading” – it “encodes a nitrate transporter”.
    2. L182 The gene expression results do not show how the loading process was affected – they show that the expression of the NRT1.5A gene was not affected.
    3. L183 atmospheric VPD DID affect NRT1.5 expression in well watered conditions.
    4. L184 The NRT1.8 gene encodes a nitrate transporter that facilitates the unloading of nitrate from the xylem…
    5. L186 no effect of VPD on gene expression
    6. L188 encodes a nitrate transporter that facilitates the distribution…
    7. L193 see above comments
    8. L202 encodes the NIR enzyme
  16. L190 “whereas a low VPD in well-watered water-stressed plants” makes no sense. The authors may have meant “whereas water stress (irrespective…”
  17. L195 but there was a significant difference in NRT1.9 under well-watered conditions
  18. The legend for Figure 4 still needs to be updated to state that the figure is showing gene expression including what the fold change is relative to. Normally fold change is used for a treatment or second timepoint relative to a first – perhaps the authors are showing fold change relative to WW high VPD conditions? It would make more sense to show fold change relative to WW low VPD conditions, as these are the conditions of acclimation/adaptation according to Figure 1. Fold change should not be shown for a factor relative to itself (currently it appears that WW high VPD is being shown relative to itself). In addition, SE should not be used for fold change values as these are not normally distributed. If the authors wish to display fold change values they need to calculate 95% confidence intervals instead.
  19. Contrary to the author’s statement in the response to reviewers and the legend, the version of Table 1 I have been given is not colour coded.
  20. L213 How was GLU affected by water availability?
  21. L219 LYS + ILE + MET + PHE + TYR do have sig differences with atmospheric VPD
  22. L245 according to Figure 1 the VPD was not increased but maintained as high.
  23. L246 Are there additional control plants that are not shown in Figure 1? If not, this sentence is confusing.
  24. In Tables S2 and S3 RcaB and TRP are statistically significant, respectively
  25. Minor typos/grammar mistakes (suggested change in square brackets):
    1. Title: Short[-term]
    2. L98 produce[d]
    3. Figure 3c contains an “e” that does not appear to correspond to anything
    4. L156 “under well-watered conditions” repeated twice in sentence
    5. Table 1 mistakenly contains bold numbers for N.
    6. L214 “On the other hand” doesn’t make sense in this context – “In addition” would be more relevant here.
    7. Inconsistent size and font in Table 1 legend.
    8. L269 asp

Author Response

Comments and Suggestions for Authors

The authors have improved the English, but unfortunately many issues with the presentation of scientific content remain, including some I previously raised. Rather than writing "the text has been revised" the authors should say where/how they revised the text to address the concer, because there are too many instances where the authors claimed to have made the change, but did not in fact make the change. The authors need to be proactive in checking that their text is consistent with the data they are presenting. Considerable changes are required before this manuscript would be acceptable for publication in any journal. The recommendations below are in order of the appearance in the text, rather than importance.

  1. Figure citations do not match figure numbers (e.g. Fig1a is actually referring to Fig2a) – this needs to be fixed throughout. The order of panels in figures should also match the order discussed in text.

Following referee 1 comment, in the revised version, Figure numbers has been corrected throughout the whole manuscript and the text matchs the order of panels in figures.

  1. Please add an in-text citation to Figure 1 (experimental) with a brief (1 or 2 sentence) explanation of experimental design at the start of the results. The explanation in the discussion could be moved up – as it would help readers to have this information before reading the results, rather than after.

Done. A brief introduction to the experimental results has been incorporated at the beginning of Results in the revised version of the manuscript (L116-120 in the revised version).

  1. It would help if the temperature regime was also shown on Figure 1.

We agree. A more detailed information have been included in Figure 1, which now shows temperature, RH and VPD data for all phases of the plants treatments.

  1. I stand by my previous comment that it would be more intuitive to have high VPD (40% RH) displayed on the right of low VPD (80%), but if the authors wish to retain this order, they should label the legend as VPD high/low instead of RH%, to match how they describe these groups in the text.

We rather prefer to maintain the order of columns in the Figures, however we have replaced high and low RH by low VPD and high VPD respectively, as suggested.

  1. Figures are still appearing in pdf form at low resolution – which is particularly a problem for Figure 4 and Figure 5 where gene names are difficult to read or illegible, respectively. Increasing text size within figures may be sufficient to fix this issue if dpi cannot be further increased.

As proposed by referee 1, text size in the Figures 4 and 5 has been increased in order to improve readability.

  1. L98 Higher AN values were found in low VPD conditions compared to high VPD conditions – the way that the sentence is currently worded implies that WS + low VPD was equal or second to WW + low VPD, whereas in reality it is ~ equal to WW + high VPD.

Referee 1 is right. It has been changed. The sentence reads now: “The lowest and highest AN values were found in the water-stressed plants subjected to high VPD and uner well-watered and low VPD conditions respectively, whereas the rest of treatments showed intermediate values (Fig. 2a).” (L123-126 in the revised version).

  1. L103 The authors still have not mentioned that gs likely did not decrease significantly with increased VPD in WS plants because stomata were already more closed than in WW.

In the revised version, we mention that stomatal conductance likely did not decrease significantly with increased VPD in water-stressed plants because stomata were already more closed than in the well-watered ones (L128-130 in the revised version).

  1. The authors are still referring to low VPD as “increasing” (e.g. L106, L149) or “decreasing” (L217) factors while Figure 1 shows that low VPD is a continuation of the VPD conditions during the adaptation period – this is thus similar to a control condition. The authors should instead either refer to the high VPD condition as “increasing” or “decreasing” factors, or the low VPD condition as being increased/decreased relative to the high VPD condition.

Sorry for this mistake, we have corrected in Figure 1 VPD conditions during acclimation and treatments periods. As can be now seen in the revised version, high VPD and low VPD correspond to treatments that increased or decreased the conditions prevailing during the acclimation phase.

  1. Figure 3 label – x axis labels are not aligned.

X axis labels have been aligned, as suggested.

  1. Significance values still appear to be mislabelled in Figure 3a (WS high VPD is shown as ab but is closer in value to c than to either a or b), Fig3c and Fig2d and Fig 2e. The authors should provide the posthoc test results to support their labelling.

Sorry, again, for the error in the labeling of these Figures. It has been corrected. Post-hoc texts are shown here for referee 1 information:

Variables

Treatments

P-value

WW High VPD

WW Low VPD

WS High VPD

WS Low VPD

δ O18

c

b

ab

a

0.001

δ C13 TOM

ab

a

ab

b

0.084

δ C13 WSC

b

a

c

c

0.0001

δ N15 non labelled

a

a

a

a

0.18

δ N15 labelled

a

b

bc

c

0.002

  1. Figure 3 significance labels are not correctly aligned.

Significance labels have been aligned in the revised version, as required.

  1. The authors still need to cite the previous studies that isolated/characterised the function of the wheat genes included in qPCR experiments in this study in the introduction/results (e.g. L156, L177) with an introduction to the wheat gene families justifying their choice of study genes. Neither of the papers that they cited in the response to reviewers mentions NRT1.3-NRT1.9. If the names of these genes have been changed, both names should be provided at least in the supplementary material.

Tong et al. (2016, cited in the unrevised version; NRT1.3) and Wang and Tsay (2011, NRT1.9) proposed these genes as coding for nitrate transporters in the leaf, possibly to the chloroplast, and in the phloem, respectively, both in Arabidopsis thaliana. The rest of the NRT genes investigated are reviewed in Dechorgnat et al. (2011, cited in the unrevised manuscript), also for Arabidopsis. As a consequence, models in Figure 5 has been changed, panels a and b. Fortunately, this modification does not affect the main conclusion of the models shown in Figure 5. A new reference for NRT1.9 gen (Wang and Tsay 2011) has been incorporated in the revised version. Text has been modified accordingly (L82-84, L202-203, L206-207, L353-354, L523 in the revised version).

New reference:
Ya-Yun Wang, Yi-Fang Tsay (2011) Arabidopsis Nitrate Transporter NRT1.9 Is Important in Phloem Nitrate Transport. Plant Cell 23 (5): 1945-1957. DOI: https://doi.org/10.1105/tpc.111.083618

  1. L158 Fig3b no longer shows a significant difference in response to increased VPD under water stress conditions, according to the author’s updated significance labels.

In the revised version, lines from L170 to L188 describe changes in C13 discrimination. Text placed in old L158 has been accordingly deleted.

  1. L181 NRT1.5A – the authors need to be consistent with their usage of gene names.

Sorry. “A” has been deleted from NRT1.5A in the revised version (L208).

  1. The authors need to be careful of their descriptions of their gene expression results – they are presenting data for gene expression levels - genes encode proteins not functions, and the predicted effect of changes in gene expression on processes is an extrapolation and this needs to be clear.
    1. L182 a gene cannot “code for nitrate loading” – it “encodes a nitrate transporter”.

The sentence has been modified. It reads now: “… the NRT1.5 gene codes for a nitrate transporter loading from root tissue into the xylem.” (L208 in the revised version).

    1. L182 The gene expression results do not show how the loading process was affected – they show that the expression of the NRT1.5A gene was not affected.

We have modified this sentence. It reads now that the expression of the NRT1.5 gene was not affected. (L208-209 in the revised version).

    1. L183 atmospheric VPD DID affect NRT1.5 expression in well watered conditions.

We agree. This is reflected in the text of L209-210 in the revised version.

    1. L184 The NRT1.8 gene encodes a nitrate transporter that facilitates the unloading of nitrate from the xylem…

The text has been modified (L210-211 in the revised version).

    1. L186 no effect of VPD on gene expression

It has been changed according to referee 1 suggestion (L345 in the revised version).

    1. L188 encodes a nitrate transporter that facilitates the distribution…

Modified (L347 in the revised version).

    1. L193 see above comments

The sentence has been modified (L350 in the revised version).

    1. L202 encodes the NIR enzyme

The text has been modified, describing first GS2 and then NIR. The comment of referee 1 has been included. (L363 in the revised versión).

  1. L190 “whereas a low VPD in well-watered water-stressed plants” makes no sense. The authors may have meant “whereas water stress (irrespective…”

Sorry, it was a mistake. It has been corrected following referee 1 suggestion (L352 in the revised versión).

  1. L195 but there was a significant difference in NRT1.9 under well-watered conditions

Text modified (L353-354 in the revised version).

  1. The legend for Figure 4 still needs to be updated to state that the figure is showing gene expression including what the fold change is relative to. Normally fold change is used for a treatment or second timepoint relative to a first – perhaps the authors are showing fold change relative to WW high VPD conditions? It would make more sense to show fold change relative to WW low VPD conditions, as these are the conditions of acclimation/adaptation according to Figure 1. Fold change should not be shown for a factor relative to itself (currently it appears that WW high VPD is being shown relative to itself). In addition, SE should not be used for fold change values as these are not normally distributed. If the authors wish to display fold change values they need to calculate 95% confidence intervals instead.

As previously explained, low VPD are not the conditions of acclimation/adaptation periods (Figure 1 has been modified accordingly). However, these conditions are closer to low VPD than high VPD, and therefore we have changed this figure showing now fold change with respect to WW and low VPD, and SE has been replaced by the 95% confidence intervals, as required.

The legend for Figure 4 has been updated as follows:

Figure 4. Effect of water irrigation (pot capacity-WW- versus 50% of pot capacity-WS) and ambient vapour pressure deficit (VPD; high versus low) on (a) Rubisco large subunit (RBCL), (b) Rubisco activase β subunit (RcaB), nitrate transporters (c) NRT1.3, (d) NRT1.5, (e) NRT1.8, (f) NRT1.4, (g) NRT1.7, (h) NRT1.9, (i) glutamine synthetase (GS2) and (j) nitrite reductase (NIR) determined on wheat flag leaves. Each value represents the fold change with respect to well-watered low VPD plants. Four replications were used. Since fold change values are not normally distributed, the 95% confidence intervals are shown.

  1. Contrary to the author’s statement in the response to reviewers and the legend, the version of Table 1 I have been given is not colour coded.

We are not sure if it is a problem of the display but, as stated in the legend “Amino acids that were significantly altered by irrigation and/or relative humidity treatments are highlighted in orange colour, whereas the amino acids that were not significantly affected are highlighted in blue colour.” We hope this will be Ok in the revised version.

  1. L213 How was GLU affected by water availability?

GLU was not affected by water availability, irrespective of the VPD conditions (L422 in the revised version).

  1. L219 LYS + ILE + MET + PHE + TYR do have sig differences with atmospheric VPD

Yes, we agree. There are differences but they are smaller than with water availability. It has been changed accordingly (L430 in the revised version).

  1. L245 according to Figure 1 the VPD was not increased but maintained as high.

Sorry again for the mistakes of Figure 1. Treatments do increase of decrease VPD (see new, corrected Figure 1).

  1. L246 Are there additional control plants that are not shown in Figure 1? If not, this sentence is confusing.

Sorry again for the mistakes of Figure 1. Treatments do increase of decrease VPD (see new, corrected Figure 1).

  1. In Tables S2 and S3 RcaB and TRP are statistically significant, respectively

Referee 1 is right. It has been corrected.

  1. Minor typos/grammar mistakes (suggested change in square brackets):
    1. Title: Short[-term]

Done. Title has been changed (L2 in the revised version).

    1. L98 produce[d]

Done (L122 in the revised version).

    1. Figure 3c contains an “e” that does not appear to correspond to anything

Right. The letter “e” has been deleted in the revised version.

    1. L156 “under well-watered conditions” repeated twice in sentence

The first “Under well-watered conditions“ has been deleted (L192 in the revised version).

    1. Table 1 mistakenly contains bold numbers for N.

It has been corrected. Sorry.

    1. L214 “On the other hand” doesn’t make sense in this context – “In addition” would be more relevant here.

“On the other hand” has been replaced by “In addition”, as suggested (L423 in the revised version).

    1. Inconsistent size and font in Table 1 legend.

Size and font in Table 1 are those used for Figure legends and text in the manuscript. In the revised version, we will check that there is no problem with font type and size.

    1. L269 asp

“asp” has been replaced by ASP (L472 in the revised version).

Reviewer 3 Report

  • At least in my version of the revised manuscript, figure text is still unreadable (Especially Figure 5). While I have no idea whether this is a problem of the embedding of the figures into the pdf, this should definitely be solved. I apologize if this is problem with my pdf reader, but please check also the converted pdf for readability before submission.

  • My comment on the unit for the amino acids in table1 has not been resolved. The numbers may be fine but the unit is not. More than 1000 mg g-1 is not possible. Last time I checked a g has 1000 mg. So how can some amino acids be over 1000? Are they maybe µg g-1? Or µmoles g-1. Also please state if these are FW or DW-related values.

  • Also amino acids are not blue, as mentioned in the table legend.

Author Response

Comments and Suggestions for Authors

  • At least in my version of the revised manuscript, figure text is still unreadable (Especially Figure 5). While I have no idea whether this is a problem of the embedding of the figures into the pdf, this should definitely be solved. I apologize if this is problem with my pdf reader, but please check also the converted pdf for readability before submission.

Text size in the Figures 4 and 5 has been increased in order to improve readability, following this comment and that of referee 1.

  • My comment on the unit for the amino acids in table1 has not been resolved. The numbers may be fine but the unit is not. More than 1000 mg g-1 is not possible. Last time I checked a g has 1000 mg. So how can some amino acids be over 1000? Are they maybe µg g-1? Or µmoles g-1. Also please state if these are FW or DW-related values.

Sorry, amino acids are expressed as µg g-1 FW. It has been corrected in the revised version.

  • Also amino acids are not blue, as mentioned in the table legend.

We are not sure if it is a problem of the display but, as stated in the legend “Amino acids that were significantly altered by irrigation and/or relative humidity treatments are highlighted in orange colour, whereas the amino acids that were not significantly affected are highlighted in blue colour.” We hope this will be Ok in the revised version.

Round 3

Reviewer 1 Report

I'm sorry to say this but I must recommend that the editor reject this paper because I no longer have faith in the data or analyses that the authors are presenting. I have now seen three versions of this paper, each containing considerable errors in terms of text not matching data, statistical groupings not matching data or being changed in every version (a simple error could be corrected once, changing the groupings on a graph twice does not engender confidence), and now data in figures being substantially changed between versions without mention or explanation. This is no longer an issue of excusable errors, where the benefit of the doubt can be given.

I suggest that the authors go back and double check all the figures that they are presenting, including statistical subgroups are an accurate reflection of their original data, and that their text accurately reflects their data before resubmitting anywhere for publication.

Author Response

As observed previously the authors would like to apologize for the potential inconveniences derived from the manuscript revision process. 

Finally the authors would like to observe that the text and figures have been revised carefully.